# Wirbelsäule-Plot for Multivariate Timeseries customized with AIP Agent

September 16, 2025

**Abstract**

Growing interest in time series analytics and forecasting leads to demand in visualization including qualitative explanatory features at point in time. Enablement of prompt-specific instructions as inputs creates new horizons in implementation of interactive charts controlled by AIP Agents. In this paper we outline new approach to visualize the multi-variate timeseries with explanatory features at point in time. Palantir Foundry is perfect platform for this task and fully equipped with AIP-powered capabilities. Combination of the AIP Agents and customizable vega chart Wirbelsäule-Plot becomes valuable resource for conducting comprehensive analysis. Our solution demonstrates interactive visualization for decision making systems.

## 1 Overview

Time series analysis and deep learning forecasting often present outputs in the form of numerical data or sentence summaries with descriptions of specific points. We propose solution of visualization the entire timeline of various historical events, enabling individuals to identify potential outliers or make informed decisions, that can help to assess the state of the study subject. We address visual solution for multi-dimensional analytics. The core idea is to visualize the multiple time series for subject of study, in which both task-specific instructions and raw time series are treated as multimodal inputs into AIP. To accomplish this goal, following distinct tasks are developed.

### 1.1 Ontology Object

Firstly, decomposition of the data source into timeline structure with main attributes Event ID, Event Type Event Label and Display Details, including other supplemental constituents, such as Color schema, descriptive content and customized labels [9]. We proceed to the time series pipeline designed to convert life events into continuous sequence of hard tokens. We apply Modality Encoding [1] and reserve field to address the gap issue and solve the inconsistency issue

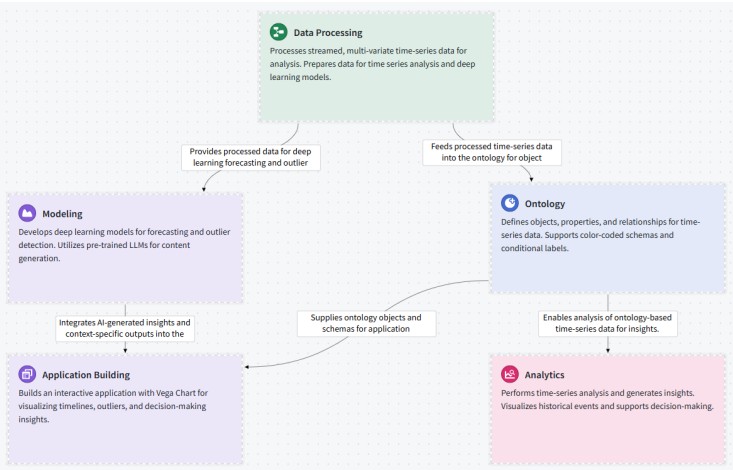

Figure 1: Data Workflow

of missing values and avoid gaps in timeline. All calendar days are labeled as predefined keyword (for example, "layoff") if no data available.

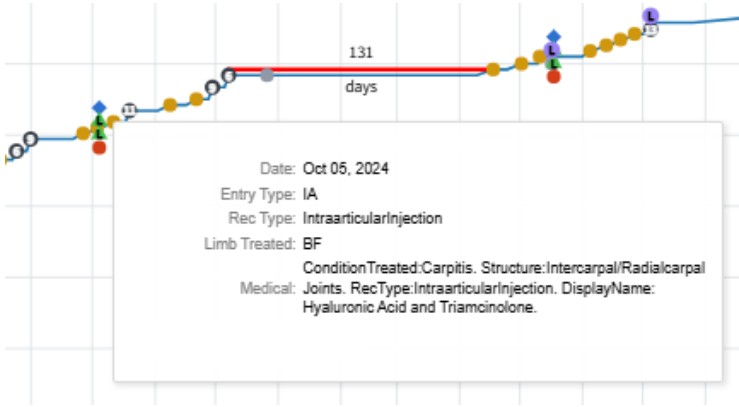

Figure 2: Multiple Events at point in time with descriptive tooltip

## 1.2 Controlled Style and Schemas

Second, each subject has classified Color of the Line, which is dynamically generated with reference to the cluster or specific class of the study group, thus each of the multiple Items per class (or event) are distinctively recognizable by color pattern. More settings are pre-generated for groups of life events. Each category has corresponding settings applied, such as shape, form or conditional label. For example, Treatment event is green triangle on the timeline of the

athlete career. Conditional labels or emoji icons can be associated with specific event at point in time. We address multiple-events that occurred on the same date with recalculated coordinate positioning against the line but within same point in time.

```
"calculate":datum.sumDistanceFurlongs + datum.yJitterUnit *
(datum.rowIndex - (datum.groupSize + 1) / 2)
```

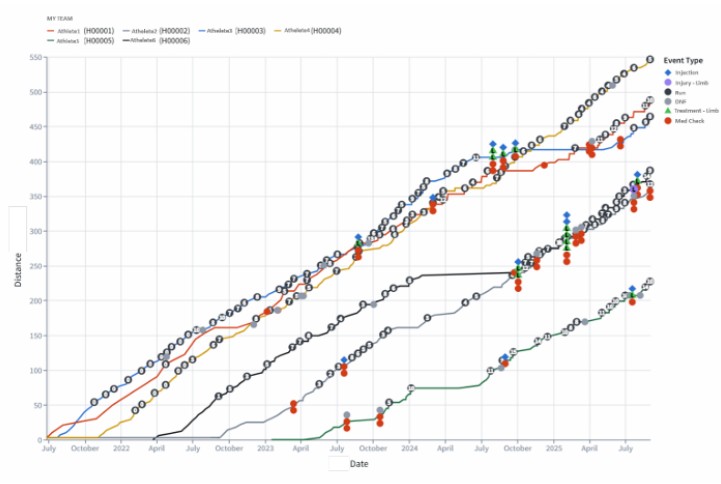

Figure 3: Distinctive Schema for each individual

## 1.3  LLM Content pre-training

Human input notes are processed through GPT 4.1 LLM to generate human readable tooltips. Thus each point in time has associated informative block, that is linked to data objects. We pre-process data to be ready in tooltip outlines at each point of the life event.[2][3]

Another block of machine learning Model objectives designed to train models for the context output to tokenize events of interest. Supervised model training, based on generic vocabularies as an input for natural language processing (NLP). We refer to Fuzzy Match BK-Tree[1] as tool for fuzzy searching in various contexts.

## 1.4  Comparison to Control groups

We employ Dynamic Time Warping (DTW)[2] algorithm for valuation of the Clusters operated by the Similarity Score.

---

[1] https://github.com/karol-broda/fuzzy-match-bktree
[2] https://pypi.org/project/fastdtw/

## 1.5 AIP Agent

AIP Agent serves users with custom data source only. Finally, we can proceed to prompt engineering in AIP Agent with configured connection to ontology objects, instructions for the json generated color and quantitative positioning schema [8]. Number of prompt experiments are conducted over AIP Agent, whose results uncover the extensive possibilities of the Vega Chart data visualization. We explore simple and flexible prompt based strategies that enable

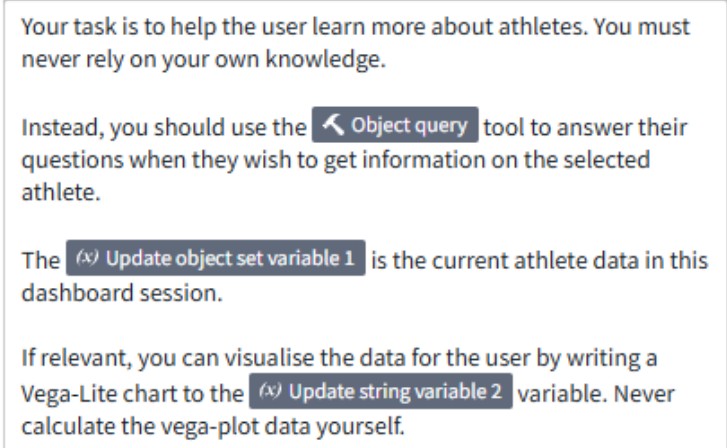

Figure 4: Prompt Construction

control over the chart component and enable interactive visualization without extensive retraining or the use of a complex external architecture. Multi-agent approach for resource optimization, where each agent has specialized prompting methods that leverage time series decomposition, patch-based tokenization, and similarity-based neighbor augmentation[1] [2] [3].

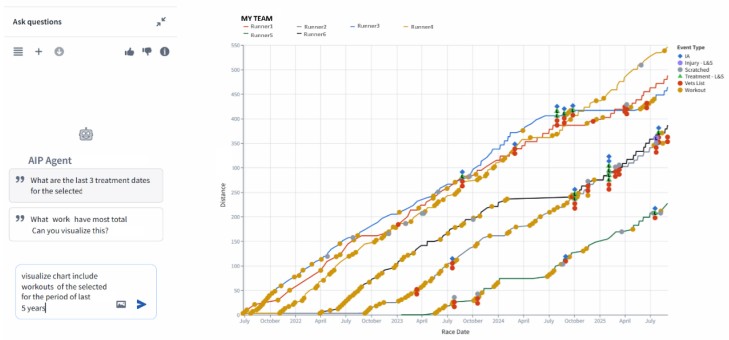

Figure 5: User request and output

We find that it is possible to enhance on-demand data retrieval while maintaining simplicity and requiring minimal preprocessing of data, that serves as interactive solution to the multi-layered timeline. To this end, we propose our parameterized Wirbelsäule Vega plot[8] component, due to it's look as a spine with numerous spinally allocated and perpendicular-positioned points.

All in all, we covered data transformation, parametrization of the view schema, multimodal LLM pre-training and prompt engineering for AIP Agents.

## 1.6 Security Policy

Privacy considerations as a top priority is handled by rules of the restricted views on Ontology Object level [3] .

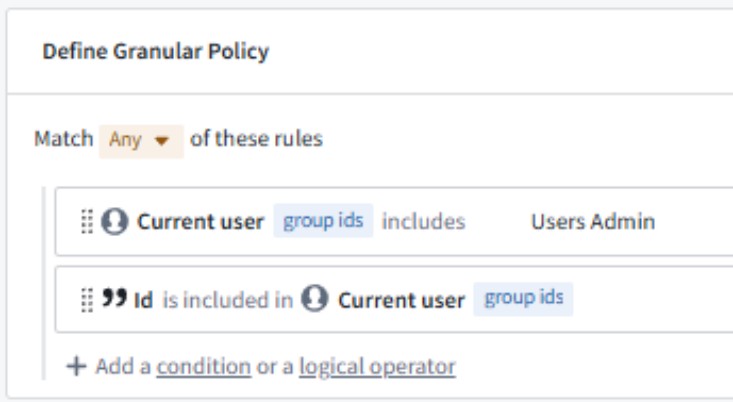

Figure 6: Granular Access Policy

Set of configurable policies based on user attributes to accommodate large groups and teams with firewall access.

### 1.6.1 Permissions on Data

Granularity is controlled by policies with complex permission tree.

### 1.6.2 Permissions on Action Types

For example, if a user meets the requirements for a policy and can see a specific data in the restricted view, then they will be able to prompt available subject.

## 2 Related Work

Institutional Athlete training considers traditional Factor Analysis of the current and past history of the athlete [4] [5]. Athlete's caretaker decision is based on

---

[3] https://www.palantir.com/docs/foundry/security/restricted-views

available evidence from medical records, and physiotherapist's speciality domain expertise and experience. This includes range of factors within categories of competition campaign, workouts, medical history, performance and placement. All in all, over 50 factors are considered and important for evaluation of the physical wellbeing.

Physiotherapist must diagnose and prevent competitive sportsmen from "bad" outcome, which can be injury, death, failed to finish competition, retirement or prolonged layoff occurred shortly after the run.

Factor Analysis approach is based on so called "cheat-sheet" with the check list of potential warning flags. Once number of potential "bad" signs is over the threshold, then specialists assigns additional check of the athlete or place on "watch list".

One picture worth thousand words, fully controlled from the AIP Agent and customizable on-request Wirbelsäule-Plot is powerful tool to answer many questions at once.

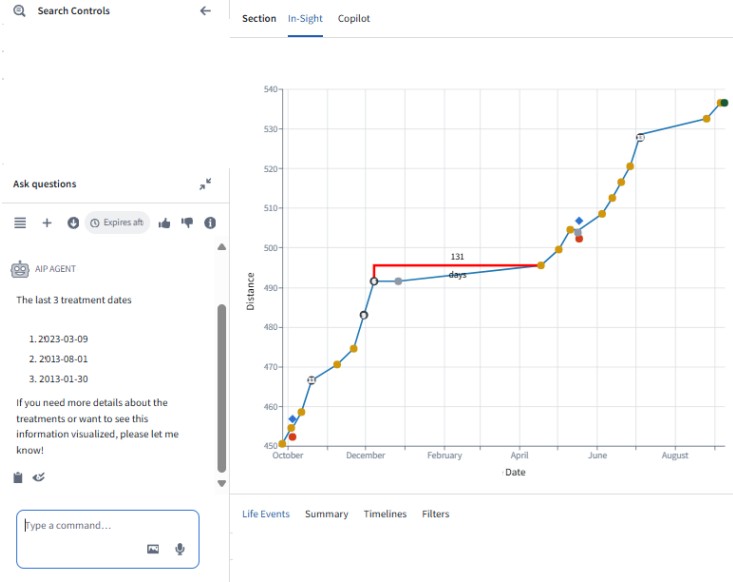

Figure 7: User request and output

We follow example of the analytics and provide example of the chart available to the Physiotherapist, who holds the right to grant permission to participate in competition, thus carefully overlook athlete's state of wellness and readiness taking into account many factors.

Visual solution provides career progress, active and career days, sport related incidents including overall standing against control groups.

Slope and pace of the workout progression provides insights on career start, pace and intensity. Abnormalities are taking into account for stress check. Another point of attention is exercise frequency so called quick turns, so that

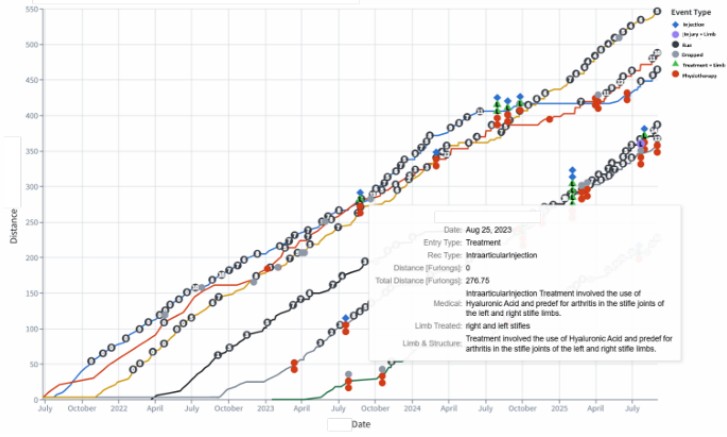

Figure 8: Control Group and Positioning of each Individual

unstable schedule can also be a warning sign. If healthcare practitioner's strategy is based on commonly used factors the quantitative score also available on the chart.

Life events are presented in variable colors and shapes, that helps to find distinctive event based on severity and can be filtered out on request from AIP Agent or traditional legend control of the checkboxes

Vega chart Wirbelsäule-Plot controlled by AIP agent with built-in a comprehensive view of operations of embedded prompts. We utilize the agent to generate tailored visualizations of the customized chart.

Wirbelsäule-Plot provides variety of the events with detailed outline. In the picture below pop-up over the Medical Treatment point provides LLM generated description based on the responsible person notes, specifically pointing area of the treatment.

# 3    Conclusion

Dynamic visualization is often a very challenging task that requires exploration and development work effort for customization. Wirbelsäule-Plot is developed to address challenges to retrieve explanatory information from multiple layers of time series. Wirbelsäule-Plot for Multivariate Timeseries customized with AIP Agent demonstrated to be part of decision making process, conducting quick answers and comprehensive visualization. As a result, "the best of two worlds" solution 1) disentangles time series embeddings, 2) AIP Agent prompts for Wirbelsäule-Plot to visualize time series events.

# 4    Limitations and Future Works

Implementation has number of limitations. First, datasets reliance human reporting and quality of notes collected by industry professionals. Second, AIP Agent relies on large data source which can be costly, thus breakdown into multi-Agent approach is required.

Next steps for development is pattern recognition, outlier identification, scenario generation and forecasted timelines.

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
