# OpenReview forum: "Wirbelsäule-Plot for Multivariate Timeseries customized with AIP Agent"
_ICLR.cc/2026/Conference — Submitted to ICLR 2026_

### Official Review · Reviewer_U6mX · 2025-10-15

**Soundness:** 1
**Presentation:** 1
**Contribution:** 1
**Rating:** 0
**Confidence:** 3

**Summary:**

This paper proposes the Wirbelsäule-Plot, a visualization framework for multivariate time series that integrates AIP Agents for interactive, prompt-based exploration and analysis. Built upon the Palantir Foundry ecosystem and Vega visualization tools, the system converts heterogeneous time-series events into multimodal, ontology-driven visual timelines.

**Strengths:**

- The proposed multi-agent AIP architecture for controlling chart parameters and prompts is technically appealing and potentially scalable, while the Vega-based implementation ensures reproducibility.
- The consideration of ontology-level security policies reflects awareness of enterprise deployment needs.

**Weaknesses:**

- The concept of "AIP Agent" remains somewhat abstract and unclear how it differs from conventional agent frameworks of LLM-driven dashboards.
- This paper does not conform to the ICLR submission format or standards and falls below the expected level of technical and scientific contribution for the venue. The paper reads more like a system description or internal technical report rather than a research paper presenting novel algorithms, theoretical insights, or rigorous evaluations. While the proposed Wirbelsäule-Plot and integration with AIP Agents may be of practical value for visualization within a specific platform, the manuscript lacks the essential components required for publication at ICLR.

Given these issues, the paper should be desk-rejected, as it does not meet the minimum scholarly, formatting, and methodological standards of ICLR.

**Questions:**

Please see the weakness above.

---

### Official Review · Reviewer_G5iu · 2025-10-29

**Soundness:** 2
**Presentation:** 1
**Contribution:** 1
**Rating:** 2
**Confidence:** 5

**Summary:**

This submission presents a method for visualizing multivariate time series using “Wirbelsäule-Plot” combined with “AIP Agents.” While interactive visualization of multivariate data is an interesting topic, the paper in its current form does not meet ICLR standards.

**Strengths:**

- The paper touches on a potentially relevant topic — visual analytics and interactive visualization for multivariate time series.

- The idea of combining visualization with language-based interaction could, in principle, be interesting if properly formulated and evaluated.

**Weaknesses:**

- The submission does not follow the ICLR paper format or academic writing conventions.

- The technical contribution is unclear and lacks methodological rigor; the proposed approach mixes unrelated concepts (AIP agents, ontology objects, GPT tooltips, etc.) without a coherent framework.

- There are no experiments, evaluations, or comparisons to prior work.

- The writing is confusing and contains vague or promotional statements (e.g., references to Palantir Foundry).

**Questions:**

- What is the concrete research problem being addressed?

- How does the proposed method differ technically from existing multivariate time-series visualization or modeling methods?

- Are there any quantitative or qualitative results that demonstrate the usefulness of the approach?

---

### Official Review · Reviewer_GBKg · 2025-10-30

**Soundness:** 1
**Presentation:** 1
**Contribution:** 1
**Rating:** 2
**Confidence:** 3

**Summary:**

The paper describes an elaboration for the plotting of multivariate time series.

The presentation is not clear. The results appear to be interesting, but the innovation is not evident.
The references are reduced. The organization should be checked. For example, the “Related Work” section is after the description of the AIP agent.
Section 1.6 Security Policy does not provide informative details in its current form.
On page 3, there is a prompt for the calculation, but it is not referred or described.
A detailed description of the AIP agent is also essential for understanding the content.

**Strengths:**

the problem is interesting, the plots show some result

**Weaknesses:**

many fundamental details are not described

**Questions:**

Which is the structure of the adopted AIP
How the AIP is trained?
How the AIP is validated?
Is the contribution based on the selection of the right prompts?
How large is the set of visualization modalities?

---

### Official Review · Reviewer_nvkk · 2025-10-31

**Soundness:** 1
**Presentation:** 1
**Contribution:** 1
**Rating:** 0
**Confidence:** 5

**Summary:**

This submission does not appear to present a complete piece of work, and the current formatting does not conform to the ICLR conference template.

**Strengths:**

None.

**Weaknesses:**

Incomplete work.

**Questions:**

None.

---

### Meta-Review · Area_Chair_h4Hm · 2025-12-26

**Summary:**

All four reviewers converge on a negative assessment and judge that the submission, in its current form, does not meet ICLR’s minimum standards. While the general topic, i.e., interactive visualization of multivariate time series with language/agent-based control, could be relevant, the manuscript reads primarily as an incomplete system description rather than a research contribution. Reviewers consistently highlight (1) non-compliance with the ICLR template and academic writing conventions, (2) an unclear problem statement and unclear technical novelty of “Wirbelsäule-Plot + AIP Agent” relative to existing visualization/LLM-dashboard approaches, (3) missing essential methodological details (agent architecture, prompt design, training/validation pipeline), and (4) the absence of experiments, evaluations, or comparisons to prior work. On this basis, I recommend rejection.

**Reviewer Concerns:**

There is no substantive author rebuttal/discussion visible in the forum thread, so the core concerns remain outstanding. In particular, the reviewers’ primary unresolved issues are the lack of a clear and rigorous research framing (what precise research question is being answered, and what is the scientific contribution), insufficient technical specification (what the AIP Agent concretely is, how it is trained/validated, and how prompt-based control is operationalized), and the lack of evidence supporting effectiveness or generality (no quantitative results, no user study/qualitative evaluation, and no baseline comparisons). In addition, multiple presentation problems were raised—ICLR formatting non-compliance, weak organization (e.g., placement of related work), unclear or unreferenced prompt/code fragments, and an uninformative “Security Policy” section—further limiting interpretability and reproducibility. Collectively, these issues prevent the work from being evaluated as a research paper at ICLR.

**Reviewer Scores:**

Given the absence of meaningful discussion/rebuttal and the fundamental nature of the concerns, I do not expect scores to move upward.

---

### Decision · Program_Chairs · 2026-01-26

Reject